# Evaluation of a Novel Nanodroplet Cutting Fluid for Diamond Turning of Optical Polymers

**DOI:** 10.3390/polym12102213

**Published:** 2020-09-27

**Authors:** Lihua Li, Hau Chung Wong, Rong Bin Lee

**Affiliations:** 1Sino-German College of Intelligent Manufacturing, Shenzhen Technology University, Shenzhen 518000, China; 2State Key Laboratory of Ultra-precision Machining Technology, The Hong Kong Polytechnic University, Hong Kong 999077, China; kero.rs.wong@connect.polyu.hk (H.C.W.); wb.lee@polyu.edu.hk (R.B.L.)

**Keywords:** diamond tool, lubrication, polymer, roughness, surface integrity

## Abstract

In this study, a novel nanodroplet cutting fluid (NDCF), consisting of emulsified water and oil nanodroplets, was developed to improve the surface quality of the single-point diamond-turned optical polymers. This developed NDCF was able to penetrate the chip–tool interface, contributing to both cooling and lubricating effects. The performance of NDCF was evaluated from perspectives of the surface irregularity, roughness, and cutting force of the machined groove in a series of taper cutting experiments. Meanwhile, a high-quality optical surface was obtained and the micro-level form error was reduced in the diamond turning of a Polymethylmethacrylate (PMMA) microlens array.

## 1. Introduction

It has been long challenging to achieve ultraprecision machining on optical grade polymers. Specifically, the chipping debris is vulnerable to melting due to the low melting points of most polymers and the high temperature generated in the heat-affected zone during machining, and consequently, its adhesion on the cutting tool would inevitably cause adhesive tool wear [1] and a poor surface finish on the machined part. In this context, it is even more difficult when it comes to ultraprecision machining using a diamond tool, which requires a nano-level surface finish and micron-level form accuracy. Most polymer-made lenses are injection-molded and produced in large quantities. However, direct machining is preferred for small quantities and the prototyping of precision optical components such as microlens arrays used in laser optics. Given the micro-level dimension of each lenslet in the microlens array, there could be thousands of lenslets over a small area. Continuous machining on them without tool changing brings about challenges to the single-point diamond turning to produce a high-quality optical surface. 

Friction reduction has been considered as the key to lowering the adhesive tool wear, and different cutting fluids have been developed to control the frictional force [2,3,4]. For example, Das et al. [5] developed a type of so-called nanoparticle-enhanced cutting fluid (NPCF) that contained suspended nano-sized particles, and its dynamic spreading in the cutting fluid was comprehensively evaluated in subsequent research [6]. Similarly, Chan et al. [7] demonstrated the capacity of nanodroplets (NDs) in significantly suppressing the thrust force vibration. Moreover, with the introduction of the taper cutting theory, Chan et al. [8] observed the reasonably well-applied volume conservation rules for cutting Al 6061 with a depth of cut (DoC) of 1–1.5 μm, which coincided with the optimal DoC range. Subsequently, material adhesion was extensively discussed [9]. As demonstrated in these previous studies, the nanodroplet cutting fluid (NDCF) is able to suppress the form error and reduce irregularities, while expanding the useable range of the DoC. However, all these studies exclusively focused on metallic materials, with no reported research on optical polymers. Accordingly, this study aims to evaluate the performance of NDCF in improving the diamond turning quality of optical polymers with a comparison to that of normal water and oil.

In this research, a novel NDCF was developed for the diamond turning of optical grade polymers, which consisted of stable nano-sized emulsified water and oil molecules. Different from nano cutting fluids containing nanoparticles [10], this NDCF was generated by breaking emulsified oil and water into stable small molecules with ultrasound energy. Moreover, it could penetrate the tool–chip interface due to its small size. Meanwhile, the high cutting pressure (often in the order of 100 MPa) at the tool–chip interface during the machining tended to prevent the conventional oil-based cutting fluids from normal operation. In this condition, the fluid served only as a coolant, instead of a lubricant. In this study, water, conventional cutting oil, and NDCF were used in a series of taper cutting experiments, with an aim to evaluate the effect of NDCF on the machined profile, surface finish, and cutting force. A single-point, round diamond tool was used to generate the microlens of required diameters. For the first time, this study investigated the effectiveness of NCDF in ultraprecision machining of optical grade polymers through evaluating the surface irregularity and roughness of the machined profile and the cutting force.

## 2. Materials and Methods 

### 2.1. An Overview of the Cutting Fluid

Organic oils have been widely used as lubricants in the machining industry, with most of them being petroleum-based. During the machining process, about 97% of the energy is transformed into heat, and the remainder is stored in the cutting tool or as residual stress in the chip formation [11]. The accumulated energy at the cutting zone generates a high temperature, which leads to tool wear, cutting edge softening, and, hence, a poor surface quality. As a countermeasure, cutting fluids are introduced to lubricate and cool the heat-affected zone, inhibit corrosion, and flush the chips away during machining, so as to enhance the surface finish and extend the tool life. Various lubrication methods have been applied in the industry, including the dry cutting method [12], the minimum quantity lubrication (MQL) method [13,14], and so on. The dry cutting method is favored because of its significant economy and environmental friendliness. However, it is not applicable to machining polymers due to the relatively poor thermal and cooling properties of air. When it comes to the MQL method, it uses lower amount of coolants (1/3–1/4 of conventional usage), which requires fluids to be highly effective in cooling and lubricating. Unfortunately, such a high effectiveness is not easy to achieve. Recently, a new class of nanofluid-based lubricants has emerged, which contains nanometer-sized particles to enhance heat transfer [15,16]. They are favored by their high effectiveness, but they are also criticized when it comes to a series of issues including environment pollution, human contact, biodegradability, oxidation, and storage stability [17]. Moreover, the abrasive nature of the nanoparticles is not suitable for machining optical grade polymers. 

In this study, a novel NDCF without nanoparticles is developed and applied to the ultraprecision diamond turning of optical polymers, followed by an examination of its performance. During the experiment, we use the following professional instruments to ensure our experimental data can be collected and analyzed with high precision and high accuracy:(1)Ultrasonic mixer (Sonxi Ultrasonic Instrument, Sonxi Inc., Shenzhen, China) for generating the NDCF(2)Sindatek Contact Angle Meter (Sindatek Inc., Taipei, Taiwan) for contact angle measurement(3)Olympus BX60 optical microscope (Olympus Inc., Tokyo, Japan) for the observation of NDCF(4)Zygo Nexview (Zygo Inc., Middlefield, MA, USA) for surface profile measurement(5)Kistler Component Dynamometer (Ksiter Inc., Winterthur, Switzerland) for cutting tool vibration measurement.

### 2.2. Preparation of the NDCF for Diamond Turning

Lubricants play an important role in lubricating and cooling the workpiece and cutting tool during the machining. Water and oil emulsions are the most commonly used lubricants in the industry. However, conventional oil-based lubricants generally fail to work well in precision machining, due to their ineffectiveness in cooling. Specifically, the diamond turning of non-ferrous alloys is generally accompanied by the employment of low-density, low-viscosity petroleum oil in single-point diamond precision machining, such as methylated spirit or kerosene, carbon tetrachloride, and compressed air sprayed through nozzles. With its high heat capacity, water works perfectly as a coolant. In comparison, oil is an ideal lubricant, but it has weak heat capacity. Moreover, oil-based lubricants are not able to penetrate the tool-workpiece interface due to their large molecules. In this context, conventional air/water-based or oil-based lubricants are all not effective in the ultraprecision machining of optical grade polymers.

There are mainly three types of cutting fluids used in the experiments—namely distilled water, emulsified oil, and NDCF. The former two were used for comparative purposes in this study. Specifically, the NDCF was produced by passing mineral oil (JAEGER SW-105) and distilled water into an ultrasonic mixer (Sonxi Ultrasonic Instrument) at a ratio of 1:1. The oil and water molecules were broken down by the high-power (600 W) ultrasonic machine which operated at a frequency of 20 kHz for 20 min. Visible light was scattered internally among the nano oil droplets, and the fluid appeared milky with some excessive oil floating to the top. During the experiment, a constant flow rate was used. The flow rate was 50 mL per hour, (0.830 mL per min.), air pressure 2 bar, nozzle distance 50 mm, spray angle 30 degrees, while the cutting area was 300 μm × 20 mm × 5 μm per min.

The NDCF was generated in the laboratory prior to the cutting experiment to ensure its freshness, and its wetting properties were assessed with a Sindatek Contact Angle Meter. Furthermore, the lubricant surface tension and energy were calculated based on the static contact angle (Figure 1a). The wetting angle of a nanodroplet was found to lie between those of water and oil drops. An optical micrograph of the NDCF is shown in Figure 1b. Several ratios of oil to water were tried and finally a 1:1 ratio was chosen for its optimal wetting properties. The size of the nanodroplet could be resolved down to hundreds of nanometers using an Olympus BX60 optical microscope at a magnification of 100× (Figure 1b). The contact angle of water is around 96.63 degrees and the one of nanodroplet is around 77.37 degrees. The biggest size of a nanodroplet is from 3.2 to 3.4 μm.

### 2.3. The Taper Cutting Experiment

A taper straight-cutting experiment was designed to evaluate the effectiveness of NDCF on the surface quality that could be obtained in the diamond turning of an optical grade polymer, herein Polymethylmethacrylate (PMMA), on a four-axis CNC ultraprecision machine Nanotech 350FG (Moore Nanotechnology System, New Hampshire, NH, USA, Figure 2a). The taper cutting experiment was chosen because it could readily reveal the effect of DoC on the machined surface roughness. Specifically, the groove was 20 mm long and 300 μm wide, with a DoC range of 0–4 μm. 

A diamond tool with a round nose radius of 2.70 mm was used to machine the PMMA (Figure 2b,c). In order to achieve smooth machining, the profile of the tool nose curvature should be replicated faithfully on the machined surface profile. However, there were inevitably geometrical deviations due to the elastic recovery of the workpiece, the tool wear, and other complicated effects. Although these deviations were very small in ultraprecision machining, they affected the quality of a high-precision optical surface. The variation of the cutting force was tracked by a transducer to assess the ability of the nanodroplets to reduce the frictional force in the machining process.

In order to have the same initial experimental conditions, the surfaces of all the specimens were first face-turned on the same CNC ultraprecision machine before the implementation of the taper cutting experiment. The workpiece was mounted in an inclined position with an angle of 0.01° to generate a taper cut (Figure 2d). The DoC was 4 μm in the beginning, and then it diminished to zero. The cutting parameters are listed in Table 1. 

## 3. Results

### 3.1. Effect of Cutting Fluids on Groove Profiles

The wear of the cutting tool was inspected and checked both before and after the experiment. After machining, the machined groove profiles and surface finish were measured on a non-contact optical profiling system Nexview (Zygo, Middlefield, MA, USA), with a measurement accuracy of up to about 1 nm. Subsequently, the measured data were analyzed with the Wyko Vision software (v3.41,Veeco, New York, NY, USA). Figure 3a shows a typical surface profile of a single groove and the optical profile scanner repeats hundreds of scans along the groove. After stitching hundreds of images, the entire machined groove profile was captured. The analysis steps are displayed in Figure 3. 

The cross-sections of grove profiles at various DoCs were extracted sequentially from the stitched surface profiles to study the nanometric surface irregularities of the groove profiles (Figure 4). It can be seen that the groove profiles cut with NDCF are the smoothest, especially in the condition of a small DoC. As shown in Figure 5, the bottom of the groove machined with NDCF was the smoothest and it was the closest to the actual tool profile. The deviation from the tool profile could be ascribed to many factors, such as micro tool wear, adhesion of the melted debris on the surface of the diamond tool [1], and the pile-up (swelling), sink-in, and elastic recovery of the deformed substrate surface at the micron-level [18,19]. Disregarding the submicron-level surface irregularities, this deviation is still undesirable for a high-precision optical lens surface. 

### 3.2. Effect of Cutting Fluids on the Surface Finish

The surface roughness was measured at DoCs of 0.5, 1.0, 1.5, and 4.0 μm, respectively (Figure 6). There are two directions along which the average roughness was measured, namely the X-axis and Y-axis (see the axis directions in Figure 2). The average roughness along the X-axis is denoted as Ra(x), that along the Y-axis is denoted as Ra(y), and the surface roughness calculated over the entire groove being cut according to the ANSI B46.1 standard is denoted as Sa.

The difference in surface roughness is small among the three cutting fluids when the DoC is below 1 μm, beyond which the difference becomes significant. The surface roughness of the PMMA cut with NDCF is often less than half of that with water or oil, and it does not increase much as DoC rises. In comparison, the surface roughness of the PMMA cut with water and oil both grows as DoC increases until 4 μm. Moreover, the PMMA machined with oil has the lowest surface quality due to it oil having the poorest performance in terms of heat absorption at the cutting zone (Figure 6a). It has been known that average roughness parameters are not able to generalize the functional nature of a surface. Another study is being conducted to correlate Ra or Sa values with the optical reflectivity of a lens surface.

### 3.3. Effect of Cutting Fluids on the Cutting Force

As recorded by a Kistler Component Dynamometer during the taper cutting (Figure 7), the cutting force signal has three components, namely the lateral force along the X-axis, the main cutting force along the Y-axis, and the thrust force along the Z-axis. Specifically, the main cutting force along the Y-axis (blue curves) decreases as the DoC along the taper groove rises. Meanwhile, the force patterns in the PMMA cut with water and oil share a high similarity. High-frequency periodic tool force vibration is observed and the jagged signal in the cutting force continues reducing until the end of cutting. In contrast, significant changes are seen in the force pattern of the PMMA cut with NDCF. The cutting force slightly drops when the cutting fluid is changed from water to oil and eventually NDCF, accompanied with an obvious reduction in the cutting force vibration. The cutting forces are relatively smooth and steady. As for the thrust force (bottom pink curves), it is smaller than the main cutting force in terms of the absolute value, and it is the smallest when NDCF is used as the cutting fluid. When it comes to the lateral force signals, they are all virtually zero. 

## 4. Discussion

Experiments were implemented to machine a test piece with 10 microlenses in a straight line, each of which had a tool radius of 0.3 mm. A microlens array on the PMMA was machined under conditions similar to those of the taper cutting experiments. A 1 mm-long straight tool path was designed, and 10 microlenses were machined continuously in a simple array. Three sets of microlenses were machined with different cutting fluids. Ten microlenses were machined on each test piece to have sufficient samples for evaluation of the surface cut with different cutting fluids.

Specifically, the employment of NDCF as the cutting fluid contributed to the smallest surface roughness along both the X-axis and Y-axis. The Sa values of the PMMA cut by water, oil, and NDCF were 20, 13, and 10 nm, respectively (Figure 8). The tool nose with a radius of 0.3 mm was faithfully replicated with micron-level accuracy using NDCF as the cutting fluid. However, this is only a preliminary test, and additional experiments are needed to machine larger arrays with a larger number of microlenses so as to enunciate the effect of the cutting fluid on the tool wear and life. More attention will be paid to explore the effect of the physical properties of NDCF on the cutting mechanism in future research. 

## 5. Conclusions

The contribution of this paper is mainly two-fold. First of all, this paper is the first report on the successful use of nanofluids in machining optical polymers. Secondly, this paper will promote research on the nano tribological aspects of the machining process and the underlying mechanism, which have not been investigated before.

Specifically, a new cutting fluid, referred to as the nanodroplet cutting fluid (NDCF), was used to machine polymers in this study, and its effectiveness was comprehensively evaluated through an integrated analysis of groove profiles, surface roughness, and force signals. NDCF was found capable of helping reduce nanosurface irregularities and improve the surface finish of diamond-turned optical polymers. Disregarding its relatively small magnitude, the improvements in both roughness and profile irregularities are of significant importance in precision optical components such as microlens arrays, in which the form accuracy and the surface finish are required to be at the micro- and nano-level, respectively. Therefore, it is safe to conclude that NDCF is a promising cutting fluid for the diamond turning of optical polymers. 

Moreover, this work can be also widely applied in the machining of nanostructures in polymeric materials, as is favored for its excellent performance in enhancing the form accuracy and surface quality. The work will also lead to further research on the tribological aspects of the cutting mechanism in the nanochip formation process. 

Furthermore, it is necessary to investigate the properties of the nanodroplets on the wear of the diamond tools and the elastic recovery of the machined surface so as to achieve a nano-level accuracy of a high-precision optical surface.

## Figures and Tables

**Figure 1 polymers-12-02213-f001:**
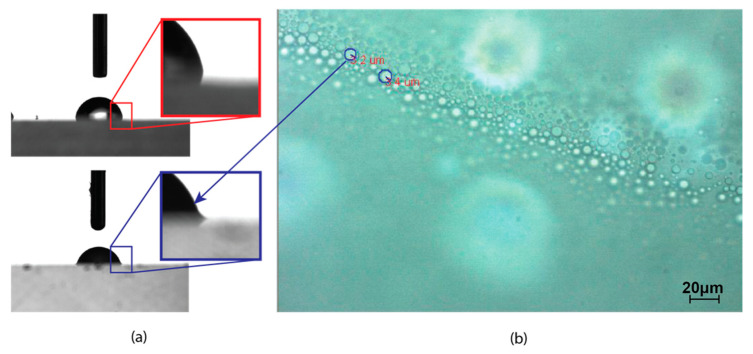
Captured images of droplets. (**a**) Contact angles of water (upper) and nanodroplet cutting fluid (NDCF) (lower); (**b**) optical micrograph of nanodroplets.

**Figure 2 polymers-12-02213-f002:**
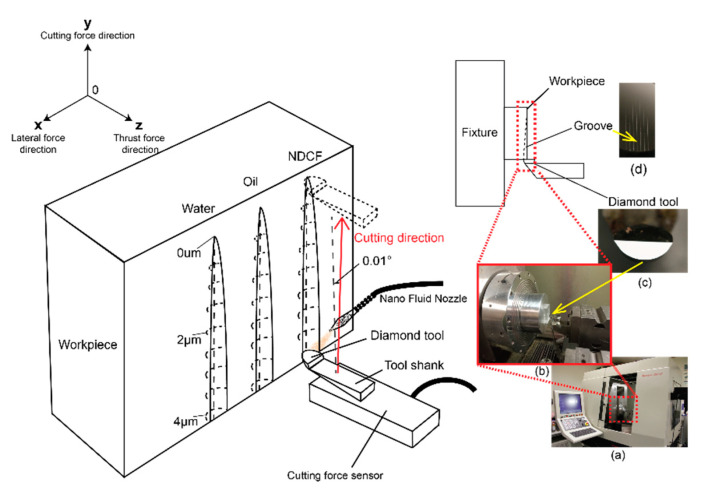
The tapper straight-cutting experiment. (**a**) Nanotech 350 CNC machine; (**b**) experiment setup; (**c**) diamond tool; (**d**) grooves.

**Figure 3 polymers-12-02213-f003:**
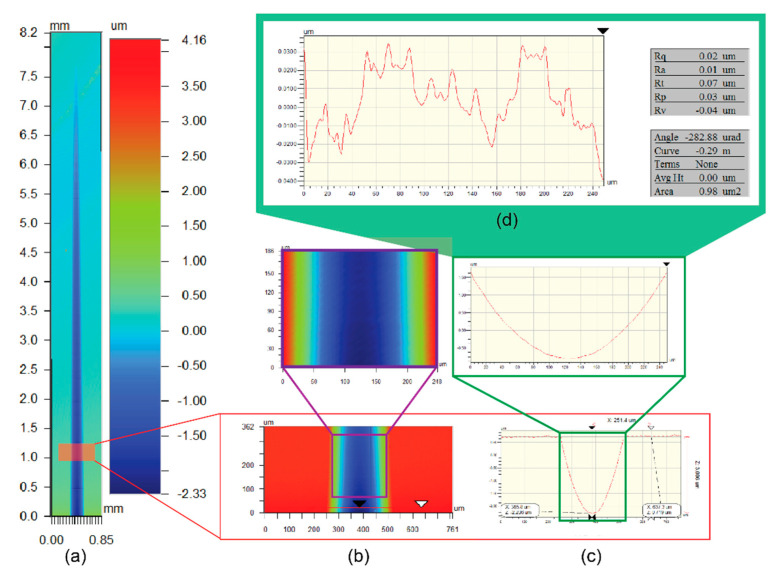
Surface data and the profile extraction of the taper cutting groove at a depth of cut (DoC) of 3 μm. (**a**) Stitched surface profile from optical profiler; (**b**) a 2D profile; (**c**) a groove profile selected for analysis; (**d**) surface roughness measured by eliminating the groove curvature.

**Figure 4 polymers-12-02213-f004:**
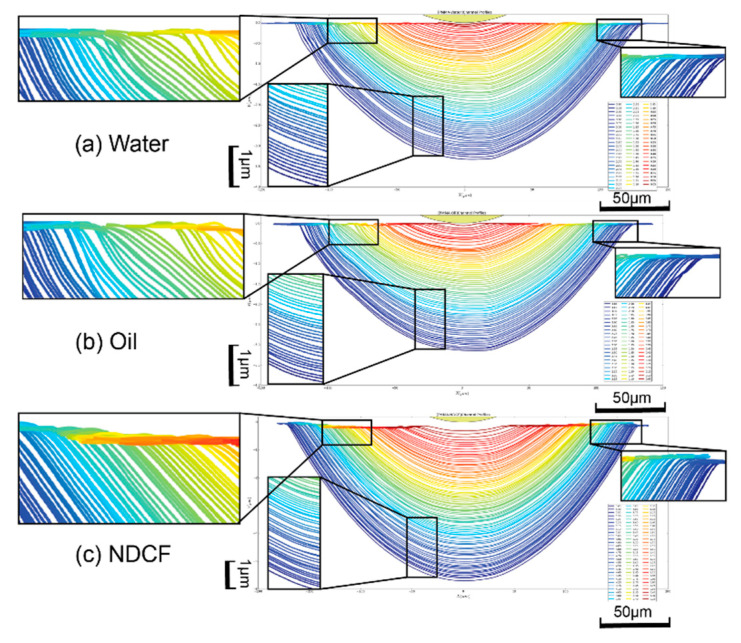
Cross-sections of grove profiles at various DoCs ranging from 0.05 to 3.30 μm with (**a**) water, (**b**) oil and (**c**) NDCF. Since DoC is much smaller than the groove diameter, the vertical scale is plotted with a higher magnification than the horizontal one.

**Figure 5 polymers-12-02213-f005:**
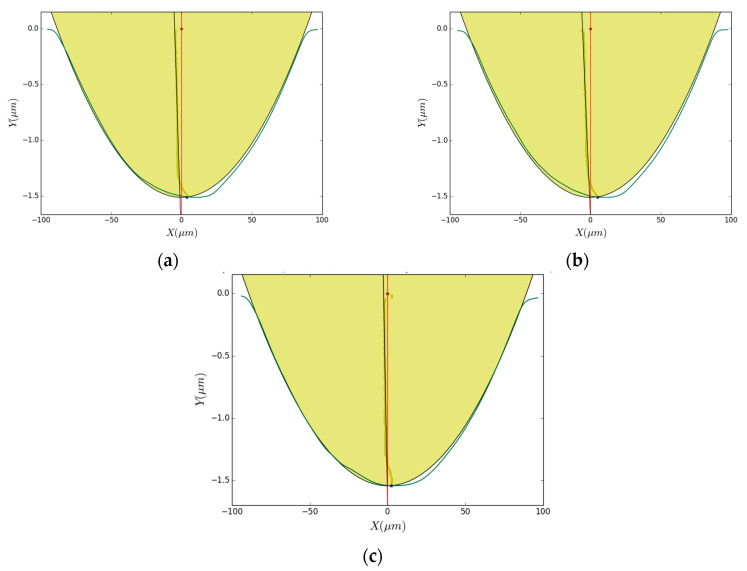
Comparison of machined groove profiles at a DoC of 1.5 μm with different cutting fluids: (**a**) water; (**b**) oil; (**c**) NDCF. The shaded part is the cross-sectional area of the diamond tool, whereas the green line is the measured profile of the machined groove.

**Figure 6 polymers-12-02213-f006:**
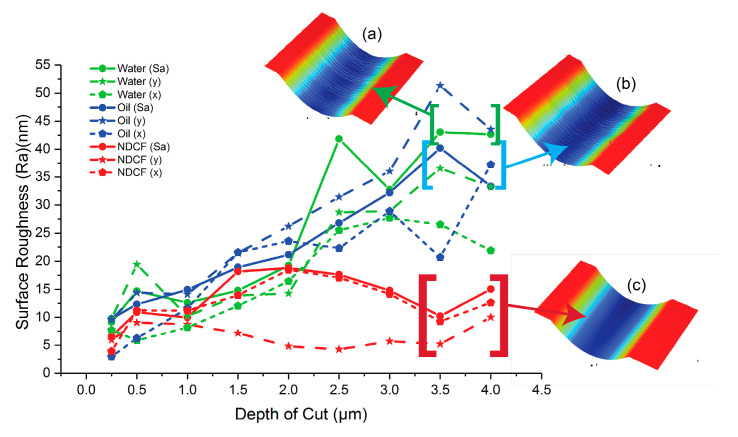
Ra(x), Ra(y), and Sa of the diamond-turned Polymethylmethacrylate (PMMA) with different cutting fluids, with samples of 2D profiles cut with (**a**) water, (**b**) oil, and (**c**) NDCF.

**Figure 7 polymers-12-02213-f007:**
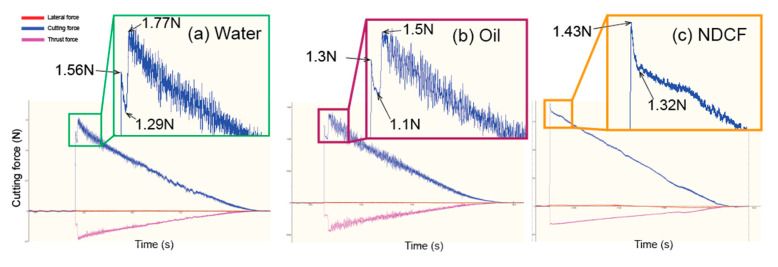
Effect of various cutting fluids on the cutting force in the taper cutting: (**a**) water; (**b**) oil; (**c**) NDCF.

**Figure 8 polymers-12-02213-f008:**
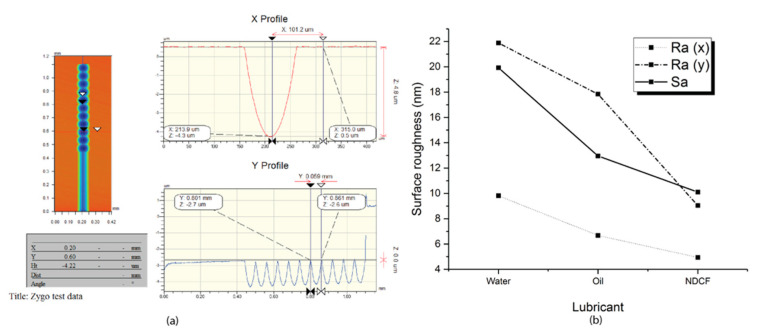
Measurements of single-column microlens arrays. (**a**) cross section profile and dimensions (**b**) surface roughness of microlens.

**Table 1 polymers-12-02213-t001:** Cutting conditions of the straight-cutting experiment.

Parameters	Value
Feed rate (mm/min)	600
DoC (μm)	0–4
Tool nose radius (mm)	2.70

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
