# Peer review of "Evaluation of a Novel Nanodroplet Cutting Fluid for Diamond Turning of Optical Polymers"

_polymers, 2020, doi:10.3390/polym12102213_

Round 1

Reviewer 1 Report

This manuscript reports the effect of the nanodroplet cutting fluid (NDCF) for diamond turning of optical polymers. Author demonstrate the advantage of NDCF compared to conventional water and oil cutting fluid, including prevention of debris adhesion and irregularities of nano-surface. Introduction part is well described about theory and advantage/disadvantage of cutting fluid examples. Therefore, I recommend this work is published in Polymers after one thing revision. In figure 2, authors should provide the two physical information for contact angle value and average size of nanodroplet and its size distribution. And Figure 1 and 2a are not need, it can replace to quantitative data.

Author Response

Response to Reviewer 1 Comments

Point 1: In figure 2, authors should provide the two physical information for contact angle value and average size of nanodroplet and its size distribution.

Response 1: The contact angle of water is around 96.63 degree and that of nanodroplet is around 77.37 degree. The biggest size of the nanodroplet is from 3.2μm to 3.4μm. We have put back these information in the revised paper.

Point 2: Figure 1 and 2a are not needed, it can replace to quantitative data.

Response 2: Thanks for your suggestion. In the revised paper, we have removed Figure 1 and 2a and added back the following descriptions:

There are mainly three types of cutting fluids used in the experiments, namely distilled water, emulsified oil, and NDCF. The former two were used for comparative purposes in this study. Specifically, the NDCF was produced by passing mineral oil (JAEGER SW-105) and distilled water into an ultrasonic mixer (Sonxi Ultrasonic Instrument) with a ratio of 1:1. The oil and water molecules were broken down by the high-power (600 W) ultrasonic machine which operated at a frequency of 20 kHz for 20 minutes. Visible light was scattered internally among the nano oil droplets, and the fluid appeared milky with some excessive oil floating to the top.  During the experiment, a constant flow rate is used. The flow rate is 50 ml per hour, (0.830 ml per min.), air pressure is 2 bar, nozzle distance is 50mm, spray angle is 30 degree, while the cutting area is 300um x 20mm x 5um per min.

The NDCF was generated in the laboratory prior to the cutting experiment to ensure its freshness, and its wetting properties were assessed with a Sindatek Contact Angle Meter. Furthermore, the lubricant surface tension and energy were calculated based on the static contact angle (Figure 1a). The wetting angle of a nanodroplet was found to lie between those of water and oil drops. An optical micrograph of the NDCF is shown in Figure 1b. Several ratios of oil to water were tried and finally a 1: 1 ratio was chosen for its optimal wetting properties. The size of the nanodroplet could be resolved down to hundreds of nanometers using an Olympus BX60 optical microscope at a magnification of 100x (Figure 1b).

Reviewer 2 Report

The paper is original and present interesting information, but is written very poorly and has not reached the level to be published in the present form. I recommend for further revision prior to acceptance. Strong side of this paper is a research character. Unfortunately a paper has a lot of mistakes (lack of information) especially in the area of tool wear and cutting tool geometry - this is important for surface roughness, cooling methods - among others, the air and liquid flows used. There is lack of discussion and references to other papers related with this topic. English can be better. Please send this article to native speaker. Below are comments concerning this paper:

  1. Introduction: it is very observational in nature. All the previous work that has been commented here it should be critically assessed (as an expert in the field) so that the research gaps can be identified. This will offer more scientific "legacy" of the work reported here. Please emphasize which research gaps the papers wishes to address.
  2. When addressing such a topic, you should add information on additives to machining fluids - you should find more information on nanoparticles, EP/AW additives etc. Please read the research of Khanna N., Maruda R., Park K.H., Leksycki K., Mia M., among others. The authors must necessarily work on the introduction.
  3. The jars shown in Fig. 1 have no scientific value. Please only leave a description of how to prepare the machining fluid.
  4. Please present the entire measuring equipment in a separate section of the paper.
  5. In the paper, which deals with cooling methods - there must be no lack of information about air flow, machining fluids flow, etc.
  6. In Fig. 7, please specify the name of the Y axis (call parameter Ra).
  7. The discussion of the results is poor. The authors try to describe what it can be seen on the graphs without any real explanation of the results. They make assumptions that are not backed-up by any references, provide the results without any theory and as a result, the conclusions are rather trivial.

  8. Conclusions are somehow simplistic as they seems to be observational without revealing findings of generic academic value. What I mean that based on the results some generic and fundamental academic conclusions need to be drawn.

Author Response

Response to Reviewer 2 Comments

Comments and Suggestions for Authors:

The paper is original and present interesting information but is written very poorly and has not reached the level to be published in the present form. I recommend for further revision prior to acceptance. Strong side of this paper is a research character. Unfortunately, a paper has a lot of mistakes (lack of information) especially in the area of tool wear and cutting tool geometry - this is important for surface roughness, cooling methods - among others, the air and liquid flows used. There is lack of discussion and references to other papers related with this topic. English can be better. Please send this article to native speaker. Below are comments concerning this paper:

Response: Thank you for your review and the useful comments.

The main purpose of this paper is to compare the performance of a nanofluid in machining optical polymers with other conventional cutting fluids such as water and oil. All the other variables are kept constant as much as possible. We used a new diamond tool for the machining. The tool was inspected under a microscope after every cut. The machining time was very short for each travel of tool and depth of cut was very small. No tool wear was found on the diamond tool. We are adding back some basic information of the tool geometry.

We are comparing the performance of oil, water and nanofluid. We have added back the flow rate of the lubricant. They were kept the same for comparison. This is the FIRST report of using nanofluid to machine optical polymers. No similar research has been reported by other researchers. We have pointed out the traditional problems of machining polymers, and also the problem of using nano-particle fluid, which is very different from the nanodroplet fluid we developed for this experiment. The nano-particle fluid is known to be very abrasive to polymers and is not recommended for cutting optical polymers. No such a comparison is needed.

We have sought the help of a technical writer from the Journal to proof read the paper.

Point 1: Introduction: it is very observational in nature. All the previous work that has been commented here it should be critically assessed (as an expert in the field) so that the research gaps can be identified. This will offer more scientific "legacy" of the work reported here. Please emphasize which research gaps the papers wishes to address.

Response 1:

The use of nanofluid as a novel cutting fluid for optical polymers is the first of its kind. There was no such previous work reported elsewhere. We are the first team to report the experiment on the machining performance of nanofluid on optical polymers. We have reviewed the use of traditional lubricants and the nanofluid is not just an improvement of an existing commercial lubricant, but a totally new one. The problems of machining polymers have been addressed in the introduction.

The new contents of the literature review added in the revised paper:

Friction reduction has been considered as the key to lowering the adhesive tool wear, and different cutting fluids have been developed to control the frictional force [2-4]. For example, Das et al. [5] developed a type of so-called nano-particle enhanced cutting fluid (NPCF) that contained suspended nano-sized particles, and its dynamic spreading in the cutting fluid was comprehensively evaluated in subsequent research [6]. Similarly, Chan et al. [7] demonstrated the capacity of nanodroplets (NDs) in significantly suppressing the thrust force vibration. Moreover, with the introduction of the taper cutting theory, Chan et al. [8] observed the reasonably well-applied volume conservation rules for cutting Al 6061 with a depth of cut (DoC) of 1-1.5 μm, which coincided with the optimal DoC range. Subsequently, material adhesion was extensively discussed [9]. As demonstrated in these previous studies, the nanodroplet cutting fluid (NDCF) is able to suppress the form error and reduce irregularities, while expanding the useable range of DoC. However, all these studies exclusively focused on metallic materials, with no reported research on optical polymers. Accordingly, this study aims to evaluate the performance of NDCF in improving the diamond turning quality of optical polymers with a comparison with that of normal water and oil.

The related references are listed as follows:

  1. Chan, C.Y., Lee, W.B. and Wang, H.. The enhancement of surface finish using water-miscible nano-cutting fluid in ultra-precision turning. International Journal of Machine Tools & Manufacture 2013, Vol. 73, pp. 62-70 .
  2. Chan, C.Y., Li, L.H., Lee W.B. ,et al. Monitoring life of diamond tool in ultra-precision machining. The International Journal of Advanced Manufacturing Technology 2016, 82, 1141–1152.
  3. Chan C Y, Li L H, Lee W B,et al. Use of Nano-Droplet Enriched Cutting Fluid (NDCF) in ultraprecision machining. The International Journal of Advanced Manufacturing Technology 2015, 84, 2047–2054

Point 2: The jars shown in Fig. 1 have no scientific value. Please only leave a description of how to prepare the machining fluid.

Response 2: We have removed it and revised the related description as follows:

The nano droplet enriched cutting fluid (NDCF) was produced by passing mineral oil (JAEGER SW-105) and distilled water into an ultrasonic mixer (Sonxi Ultrasonic Instrument) in a ratio of 1:1. The oil and water molecules were broken down by the high power ultrasonic machine which operated at 20KHz and output power 600W for 20 minutes. Visible light is scattered internally among the nano oil droplets, and the fluid appears milky with some excessive oil floating to the top which is skimmed off.

Point 3: Please present the entire measuring equipment in a separate section of the paper.

Response 3: According to your suggestion, we have added the following in the revised paper:

During the experiment, we use the following professional instruments to keep our experimental data can be collected and analyzed with high precision and high accuracy:

1) ultrasonic mixer (Sonxi Ultrasonic Instrument) for generating the NDCF

2) Sindatek Contact Angle Meter for contact angle measurement

3) Olympus BX60 optical microscope for the observation of NDCF

4) Zygo Nexview for surface profile measurement

5) Kistler Component Dynamometer for cutting tool vibration measurement

Point 4: In the paper, which deals with cooling methods - there must be no lack of information about air flow, machining fluids flow, etc.

Response 4: As the purpose of the paper is to compare the performance of the nanofluid with water and oil, a constant flow rate is used. The flow rate is 50 ml per hour, (0.830 ml per min.), air pressure is 2 bar, nozzle distance is 50mm, spray angle is 30 degree, while the cutting area is 300 um x 20mm x 5um per min. We have added the related information in the revised paper.

Point 5: In Fig. 7, please specify the name of the Y axis (call parameter Ra).

Response 5:. The Fig. has been revised as follows:

Point 6: The discussion of the results is poor. The authors try to describe what it can be seen on the graphs without any real explanation of the results. They make assumptions that are not backed-up by any references, provide the results without any theory and as a result, the conclusions are rather trivial.

Response 6:

As said, this is the first experimental report on the outcome of using nanofluid as a cutting fluid for machining optical polymers. The results show significant improvement in machining optical polymers in applications such as micro-lens arrays (MLA), which is a great success in itself.

The tribology of machining at the nano regime is very complicated. There is no theory which can explain the machining outcome. This will warrant the next phase of research on the underlying mechanism.

Point 7: Conclusions are somehow simplistic as they seems to be observational without revealing findings of generic academic value. What I mean that based on the results some generic and fundamental academic conclusions need to be drawn.

Response 7: The conclusion has been revised as follows:

The contribution of this paper is mainly twofold. First of all, this paper is the first report on the successful use of nanofluids in machining optical polymers. Secondly, this paper will promote research on the nano tribological aspects of the machining process and the underlying mechanism, which have not been investigated before.

Specifically, a new cutting fluid, referred to as the nanodroplet cutting fluid (NDCF), was used to machine polymers in this study, and its effectiveness was comprehensively evaluated through an integrated analysis of groove profiles, surface roughness, and force signals. NDCF was found capable of helping reduce nano-surface irregularities and improve the surface finish of diamond-turned optical polymers. Disregarding its relatively small magnitude, the improvements in both roughness and profile irregularities are of significant importance in precision optical components such as microlens arrays, in which the form accuracy and the surface finish are required to be micro- and nano-level, respectively. Therefore, it is safe to conclude that NDCF is a promising cutting fluid for the diamond turning of optical polymers.

Moreover, this work can be also widely applied in the machining of nanostructures in polymeric materials, as is favored by its excellent performance in enhancing the form accuracy and surface quality. The work will also lead to further research work on the tribological aspects of the cutting mechanism in the nanochip formation process.

Furthermore, it is necessary to investigate the properties of the nanodroplets on the wear of the diamond tools and the elastic recovery of the machined surface so as to achieve the nano-level accuracy of a high-precision optical surface.

Round 2

Reviewer 2 Report

The paper is original and present interesting information. Strong side of this paper is a research character and idea. The authors have considered all the suggestions. The paper may be published in its current form.